# Effect of Nitrogen Arc Discharge Plasma Treatment on Physicochemical Properties and Biocompatibility of PLA-Based Scaffolds

**DOI:** 10.3390/polym15163381

**Published:** 2023-08-11

**Authors:** Olesya A. Laput, Irina V. Vasenina, Alena G. Korzhova, Anastasia A. Bryuzgina, Ulyana V. Khomutova, Sitora G. Tuyakova, Yuriy H. Akhmadeev, Vladimir V. Shugurov, Evgeny N. Bolbasov, Sergei I. Tverdokhlebov, Aleksandr V. Chernyavskii, Irina A. Kurzina

**Affiliations:** 1Chemical Department, National Research Tomsk State University, 36 Lenin Ave., Tomsk 634050, Russia; korzhova17@gmail.com (A.G.K.); bryuzgina2016@mail.ru (A.A.B.); ugoroshkinau@gmail.com (U.V.K.); tuyakova123@gmail.com (S.G.T.); kurzina99@mail.ru (I.A.K.); 2P.N. Lebedev Physical Institute, 53 Leninsky Prospekt, Moscow 119333, Russia; ivpuhova@mail.ru; 3Institute of High Current Electronics, 2/3 Akademichesky Ave., Tomsk 634055, Russia; ahmadeev@opee.hcei.tsc.ru (Y.H.A.); shugurov@opee.hcei.tsc.ru (V.V.S.); 4Scientific and Educational Center Named after B.P. Weinberg, National Research Tomsk Polytechnic University, 30 Lenin Ave., Tomsk 634050, Russia; ebolbasov@gmail.com (E.N.B.); tverd@tpu.ru (S.I.T.); 5Nanocenter MIREA, MIREA—Russian Technological University, 78 Vernadskogo Ave., Moscow 119454, Russia; chernav@yahoo.com

**Keywords:** polylactic acid, low-temperature plasma, arc discharge, chemical compound, wettability, free surface energy, cell viability

## Abstract

The effect of low-temperature arc discharge plasma treatment in a nitrogen atmosphere on the modification of the physicochemical properties of PLA-based scaffolds was studied. In addition, the cellular-mediated immune response when macrophages of three donors interact with the modified surfaces of PLA-based scaffolds was investigated. PLA surface carbonization, accompanied by a carbon atomic concentration increase, was revealed to occur because of plasma treatment. Nitrogen plasma significantly influenced the PLA wettability characteristics, namely, the hydrophilicity and lipophilicity were improved, as well as the surface energy being raised. The viability of cells in the presence of the plasma-modified PLA scaffolds was evaluated to be higher than that of the initial cells.

## 1. Introduction

The number of skin defects constantly increasing due to burns, injuries, and surgical interventions has aroused great interest in the scientific community toward the development of medical materials intended for damaged skin repair [1]. So-called “wound dressing” includes not only textile materials but also powders, films, sponges, gels, and combinations of various components. Such materials cure by cleansing the wounds and creating optimal conditions for regeneration [2].

Superficial skin wounds affect the epidermal layer, while deeper wounds can damage the integrity of the underlying dermis. The dermis is a connective tissue composed of extracellular matrix fibers and cells. For the treatment of skin wounds, it is necessary to select a medical material that will recreate the architecture of the damaged tissue and promote regeneration. The wound dressing should have properties that are similar to those of human skin. Currently, more and more studies are aimed at creating materials that are as close as possible to the skin in terms of properties and structure. The variety of created wound coverings is explained by the large number of polymers used. The main requirements for polymeric materials to be used in biomedicine are anti-allergenicity, biocompatibility, and biodegradability; moreover, the polymer and its degradation products should not cause toxic effects.

At present, this problem is solved by producing individual (personalized) implants combined with bioactive substances. They are scaffolds based on synthetic bioresorbable materials enriched with cells belonging to a specific person. As a result, scaffolds correspond to the damaged area with great accuracy in terms of structural and biomechanical features and do not have a negative immunological reaction [3]. Scaffolds maintain tissue integrity and provide a substrate layer for the adhesion, migration, and proliferation of cells involved in the regeneration process [4]. In the medical industry, scaffolds must have the following properties: (1) high porosity; (2) rigidity corresponding to the mechanical properties of body tissues, as this feature affects cell differentiation; (3) lack of toxicity; and (4) biodegradability [5]. The degradation rate of the matrix material should approximately correspond to the growth rate of new tissue. To obtain such a matrix with the required properties, one must competently choose materials and production methods that are appropriate for creating scaffolds. 

The most suitable biodegradable material for medical applications is polylactic acid (PLA) since its degradation products are nontoxic to the human body. PLA is a biocompatible, thermoplastic aliphatic polyester whose monomer is l-lactide. A distinctive feature of this material is its ability to decompose due to the presence of ester bonds in its molecular structure. PLA degrades in the body into lactic acid molecules, which are further involved in biochemical reactions.

There are various ways to obtain medical scaffolds with desired properties. One of the most promising methods is electrospinning [6]. Electrospinning is the process of obtaining fibers from polymer solutions under the influence of an electrostatic field [7]. Depending on the characteristics of the polymer solution, environmental conditions, and device parameters, one can control the structure and properties of the product. This method is convenient to use and is suitable for producing thin fibers with a diameter from a few nanometers to a millimeter [8]. PLA-based scaffolds obtained via electrospinning meet the requirements for medical devices used as wound dressings. However, the disadvantages of PLA-based materials include the low wettability of the surfaces as well as the absence of nitrogen-containing functional groups in their surface layer, which ensure the most efficient colonization and growth of the human cell population [9].

These problems can be solved by modifying the surface with low-temperature plasma treatment in a nitrogen atmosphere. The advantages of the plasma modification of polymers include the possibility of processing only the surface and thin near-surface layers of the polymer without changing the properties of the main mass of the material [10,11]. Plasma treatment makes it possible to implement the following: the modification of the chemical and elemental composition of the surface layer of a material, the formation of a given relief on the surface, surface structure change, etc. [12]. This is possible due to both the cleaning of the surface from contaminants and the appearance of new chemical bonds [13]. 

Various types of discharges can be used to form plasma: glow, barrier, arc, and others [14,15,16,17]. It has been noticed that glow discharge for plasma formation has some limitations. Namely, a precursor is often needed to obtain a new nitrogen-containing chemical bond and for the chemical bonding of residual oxygen-forming oxides on the surface in the process of ion plasma nitriding [18,19]. This requirement complicates the operation of devices based on this type of discharge since it increases environmental and safety requirements for production and slows down the process of plasma treatment. In turn, when the barrier discharge plasma is generated, streamers are formed that can penetrate a target made of a polymer material, leaving holes [20]. The use of arc discharge can reduce the processing time and provide a more gentle mode of polymer surface treatment, while the introduction of a precursor for the formation of a nitrogen-containing bond is not required.

The purpose of this work was to study the effect of changes in the physicochemical characteristics of PLA-based scaffolds happening due to nitrogen arc discharge plasma treatment on the primary cytotoxicity of the material.

## 2. Materials and Methods

### 2.1. Obtainment of PLA-Based Scaffolds

PLA for electrospinning and further plasma treatment was obtained from the l-lactide initial monomer, synthesized in the laboratory of polymers and composite materials of NR TSU according to the technique described in [21]. The average molecular weight of the obtained PLA was 150,000 g/mol. Chloroform was used to obtain a spinning PLA-based solution. The scaffolds were produced in the Weinberg Research Center of Tomsk Polytechnic University (TPU) using the Nanon-01 electrospinning setup (MECC Co., Fukuoka, Japan). The scaffold was formed using a cylindric collector of 200 mm in diameter at an average temperature of 23 °C in the chamber, and the relative humidity was φ = 15%. The experimental setup scheme is presented in Figure 1. 

The basic process parameters are shown in Table 1. During preparation, the PLA canvas was cut with a scalpel along the collector rotation axis, and then the material was removed from the collector. Before treatment with plasma, the obtained scaffolds were loaded into a vacuum chamber at a pressure of 10^−2^ Pa for 10 h at T = 100 °C to remove residual solvents. 

### 2.2. Plasma Treatment of PLA-Based Scaffolds

The surface treatment of the PLA scaffolds was carried out in the COMPLEX setup (developed by the Laboratory of Plasma Emission Electronics of HCEI SB RAS) by using a PINK thermionic cathode plasma source in a nitrogen atmosphere [22]. The PLA scaffold samples were placed on a glass slide, 150 mm from the outlet of the PINK gas discharge plasma generator based on a non-self-sustaining arc discharge with heated and hollow cathodes. The working pressure of 0.3 Pa was maintained with fore-vacuum and turbomolecular pumps. The discharge current was set at 5 A at a filament current of 120 A and a discharge power of 250 W. The processing time varied from 5 to 30 min. The discharge of the PINK plasma generator burned diffusely in the entire volume of the vacuum chamber without a directed flow. No electrical bias voltage was applied to the samples. The concentration of plasma was ≈10^10^ cm^−3^, and the electron temperature was ≈1–3 eV. Figure 2 shows the photos of the PINK plasma generator as part of the laboratory setup for the plasma modification of solid surfaces. Table 2 shows the surface modification parameters of the PLA-based scaffolds in the nitrogen atmosphere using the PINK plasma generator.

### 2.3. Investigation Techniques

#### 2.3.1. Chemical Composition

The surface elemental composition was studied using X-ray photoelectron spectroscopy (XPS) using a PHIX-tool automated XPS microprobe with a KαAl source. The monochromated X-ray source was applied for XPS analysis. The 400 µm^2^ X-ray spot was used. A standard charge compensation system with low energy of electrons and ions (≈0.1 eV) was exploited during the analysis. The Casa XPS (CasaXPS Version 2.3.25, Casa Software Ltd., Teignmouth, UK) software was used for data analysis. The calculation of the atomic content of elements was carried out using the characteristic factors of relative sensitivity. To study the chemical environment of atoms on the surfaces of the PLA samples, the high-resolution spectra were deconvoluted. The approximating line shapes, peak half-widths, and binding energies corresponding to the peak maximum were determined from the spectra of the PLA samples. The Tougaard background was used for noise subtraction. To construct a mathematical model of the spectra, the Gaussian–Lorentzian function was applied to the elementary components.

#### 2.3.2. Wettability

Ethylene glycol and glycerol contact angles were measured using a sessile drop technique using a Kruss Easy Drop (DSA25) instrument (KRÜSS, Hamburg, Germany). The surface energy calculation was carried out using the Owens–Wendt equation [23].

#### 2.3.3. Surface Morphology

The surface morphology was studied using a Quanta 200 3D scanning electron microscope (SEM) and focused ion beam (FIB) instrument (Hillsborough, OR, USA). Magnifications from 1000 to 15,000 were used at an accelerating voltage of 20 kV. Before the SEM study, the samples were coated with a conducting gold film of a 2–5 nm thickness using magnetron sputtering to alleviate charge buildup on the surfaces. The surface roughness was studied using atomic force microscopy (AFM) using an NTEGRA Aura (Moscow, Russia) scanning probe microscope in tapping mode. The NSG01 probe of NT-MDT Spectrum Instruments (Moscow, Russia) with a resonant frequency of 150 kHz and force constant of 5.1 N/m was employed, and the scanning area was 50 × 50 μm^2^. For the analysis, the Gwyddion software (Version 2.62) was used. After the survey, XY autoplane correction was performed.

#### 2.3.4. Monocyte Isolation and Culture

Monocytes were isolated from the buffy coats of healthy donors [24]. The buffy coats were obtained from the Blood Transfusion Department of the Northern Clinical Hospital (Seversk, Russia). The obtained monocytes were cultured at a concentration of 10^6^ cells/mL in the X-VIVO 10 medium (Lonza, Verviers, Belgium) supplemented with 1 ng/mL of M-CSF (Peprotech, Hamburg, Germany) and 10^−8^ M of dexamethasone (Sigma-Aldrich, Darmstadt, Germany). The scaffolds were cut into square-shaped pieces of 0.7 cm^2^ in size, which were later UV-sterilized. 

#### 2.3.5. Alamar Blue Assay

The viability of the cells was assessed after cultivation (the evaluation of the metabolic activity of the cells). The Alamar Blue fluorescent assay (Sigma, St. Louis, MO, USA) allowed us to assess the cell viability of the non-stimulated monocytes cultured with the materials on day 6. To perform the analysis, 0.5 mL of the cell culture medium was left in each well, and Alamar Blue was added (the Alamar Blue/medium ratio was 1/10). The cells cultured without the materials served as a positive control. The X-VIVO serum-free medium supplemented with Alamar Blue was used as a negative control (without living cells). The cells were incubated together with the Alamar Blue for 3 h at 37 °C in the CO_2_ incubator. Then, the supernatants were collected into a 96-well plate, and the absorption intensity (determined at 570 nm and 600 nm) was analyzed using a Tecan Infinite 200 microplate reader (Tecan, Männedorf, Switzerland). All the measurements were performed in triplicate.

#### 2.3.6. Statistics

The data were subjected to statistical analysis using a two-tailed Student’s *t*-test. *p*-values of <0.05 were considered statistically significant.

## 3. Results

### 3.1. Surface Chemical Composition

Figure 3 shows the regions of the photoelectron spectra corresponding to the shells of atoms, which were used to calculate the elemental composition. To calculate the atomic content of carbon, the C1s region with a relative sensitivity factor of 1.00 was used. To calculate the atomic content of oxygen, the O1s region with a relative sensitivity factor of 2.93 was used. To calculate the atomic content of nitrogen, the N1s region with a relative sensitivity factor of 1.80 was used. These spectra revealed the presence of the main PLA elements carbon and oxygen, but a new peak with a binding energy of ∼399.9 eV, corresponding to atomic nitrogen, appeared following nitrogen plasma treatment. The maximum atomic nitrogen content (25.15 ± 0.3 at.%) was observed during the plasma treatment of the PLA surface for 10 min. We found that the carbon atomic concentration increased after plasma modification because of the reduction in the atomic oxygen concentration (Table 3). The atomic [C, at.%]/[O, at.%] ratio increased with nitrogen plasma treatment, indicating surface carbonization. We also found that the atomic [C, at.%]/[O, at.%] ratio after plasma modification increased from 2.00 to 2.37. This confirms our suggestion regarding the surface carbonization process during plasma treatment.

In accordance with the structural formula of the initial PLA, the mathematical model was built using three chemical environments of carbon atoms. Figure 4 represents the C1s spectra of PLA in its initial state (Figure 4a) and after nitrogen plasma treatment of different durations (Figure 4b–d). The initial C1s spectrum of PLA contains the characteristic lines of the material that correspond to the reference data on the binding energy of the electrons located at the corresponding levels of carbon in PLA [25]. The rearrangement of the ratio of the characteristic chemical bonds in PLA, as well as the new bond formation, occurred after surface modification (Table 4).

Moreover, the content of carbon bonds in the -O-CH- (2) and O-C=O (3) coordinations decreased after plasma treatment. This phenomenon may be associated with the processes of decarbonylation and decarboylation in macromolecules [26]. The highest decrease in the content of carbon bonds in the O-C=O (3) coordination by a factor of 3.36 with respect to the initial state was observed during the nitrogen plasma treatment of the PLA surface for 28 min. The minimum content of carbon bonds in the -O-CH- (2) coordination equal to 8.93 at% corresponded to the 5 min plasma treatment (Table 4). The subsequent increase in this bond content during the processing time enhancement may have resulted from the formation of a number of cross-links between macromolecules in the modified layer. The content of carbon bonds in the CH_3_-CH- (1) coordination was shown to increase after plasma treatment. The maximum content of CH_3_-CH- equal to 51.67 at.% was observed when the PLA surface was treated with nitrogen plasma for 10 min. When the processing time increased to 20 and 30 min, the contents of this bond were 47.78 and 46.66 at.%, respectively. The content increase in carbon bonds in the CH_3_-CH- (1) coordination in relation to the initial state indicates that the PLA surface layer was carbonized in the plasma treatment conditions. This carbonization process probably had a cumulative effect, lasting for a definite processing time (~10 min), followed by the rupture of the polymer chains when the treatment duration increased. 

Due to the formation of nitrogen atoms on the PLA surface (according to the results of the survey XPS spectra), elementary components corresponding to the most probable chemical environments of carbon atoms formed during the plasma treatment were introduced to fit the mathematical model. According to the C1s spectra, treatment of the PLA surface with arc discharge nitrogen plasma resulted in the destruction of the polymer chains accompanied by -C-N bond formation with a binding energy of 286.4 eV [27,28]. The maximum atomic concentration of this bond (15.20 at.%) was observed during the plasma treatment of the PLA surface for 5 min. And the subsequent enhancement of the processing time decreased the -C-N bond content to 12.59 at.% (Table 4). The N1s spectra confirm the -C-N bond formation as well (Figure 5). The total atomic concentration of nitrogen deposited on the PLA surface changed in the same way, which probably resulted from the heating and ablation of the material on the polymer surface when processing for more than 5 min. 

### 3.2. Wettability of PLA Scaffolds

Wettability plays an important role in the biochemical processes that occur on the ostein–liquid border in living organisms. The contact angle of the PLA surface was measured using the sessile drop technique for two liquids: the polar one was water, and the dispersive one was glycerol. Wetting, as a phenomenon occurring when a solid body surface is in contact with a liquid, is characterized by a contact angle *θ*, whose vertex lies at the three-phase contact point, with one side being the solid–liquid surface and the other side being the tangent to the surface of the wetting liquid [29]. The initial PLA scaffold contact angles were 121.2° with water and 122.7° with glycerol. Figure 6 shows that the PLA surface modification with arc discharge nitrogen plasma treatment led to a significant decrease in the water and glycerol contact angles. The values decreased to much less than 90° (~20° for both liquids); hence, the surface became hydro- and oleophilic, respectively.

The PLA wettability data reveal a tendency to free surface energy enhancement accompanied by a simultaneous increase in the surface polar and dispersive parts after nitrogen plasma treatment. The maximum surface energy value of 76.53 mN/m corresponds to the PLA scaffolds treated with nitrogen plasma for 5 min (Figure 7). 

### 3.3. Microstructures of PLA Scaffolds

The surface morphology changes in the PLA scaffolds after plasma treatment are shown in Figure 8. The SEM image of the untreated PLA (Figure 8a) demonstrates the regular structures of individual fibers with a characteristic electrospinning-conditioned texture.

After the plasma treatment lasting for 5 min, one can see that the fiber structure of the scaffolds is preserved, while the middle diameter of fibers (d_mid_) is increased from 0.84 μm to 4.86 μm. The fibers seem to be flattened because of the “relaxation” of the fibers under local temperature exposure during the plasma treatment. The exposure time enhancement to 10 min resulted in local damage to the PLA fibers. The plasma treatment lasting for 20 min deepened the fiber destruction. After the 30 min plasma exposure of the PLA scaffolds, in addition to the destruction process, fiber sintering took place, which limited the use of these modified scaffolds.

An increase in the surface roughness of the PLA samples with an increase in the plasma exposure time was established (Table 5). The maximum increase in roughness *R_a_* was observed for the 30 min plasma treatment PLA sample (from 15 to 248 μm); this effect is probably associated with the destruction of individual fibers of the material due to thermal exposure.

### 3.4. Effect of PLA Scaffolds on Macrophage Viability

We investigated the effect of PLA scaffolds on cell death. The main clinical problem of implant application is inflammation, both acute and chronic. Tissue macrophages are key cells capable of both stimulating and suppressing inflammatory responses in the implant microenvironment. We used a model system to study the influence of biodegradable scaffolds on primary human macrophages.

Figure 9 represents the experimental results. We found that the cell viability in the presence of the initial PLA was less than that of the control, but it was much greater than that of the negative control. In the presence of PLA + N_2_ plasma treatment lasting for 5 min, the macrophage viability of Donor 1 and Donor 2 increased to 101% and 102%, respectively. The cell viability of Donor 3 in this case was higher than that in the presence of the initial PLA. The increased viability compared with that of the initial PLA was observed for the cells of Donor 1 and Donor 2 in the presence of PLA + N_2_ plasma treatment lasting for 10 min. The negative control value was the background signal and was subtracted from the results.

## 4. Discussion

The changes in the chemical and elemental compositions of the PLA scaffolds caused by plasma treatment in a nitrogen atmosphere influenced their wettability characteristics. It was shown that the increasing nitrogen atomic concentration resulted in the water contact angle decreasing. The maximum atomic nitrogen concentration of 25.15 at.% in the PLA surface layer after 10 min of plasma treatment corresponded to the minimal value of the water contact angle equal to 15.4°. However, when the atomic concentration of nitrogen during the 30 min plasma modification decreased, the contact angle value increased again but still remained lower than the initial angle (Figure 10).

Glycerol is a liquid that has less polarity than water, so the dispersion component of the surface tension of this liquid is more susceptible to change. The atomic ratio of [C, at.%]/[N, at.%] was found to influence the wettability alteration of the plasma-treated PLA scaffolds upon contact with glycerol. The contact angles of the PLA scaffolds decreased when the [C, at.%]/[N, at.%] ratio increased. The minimum value of the contact angle (21.5°) corresponded to the maximum [C, at.%]/[N, at.%] ratio of 3.65 at.% when the PLA was treated with nitrogen plasma for 30 min (Figure 11, Table 3).

We have revealed the dependence of the wettability alteration on the chemical compounds in the PLA scaffolds, specifically -C-N bond formation. The maximum content of -C-N bonds corresponded to the highest value of the surface energy. Concomitantly, the increase in the plasma processing time linearly decreased the atomic concentration of nitrogen-containing bonds and the free surface energy as well. The maximum value of the atomic -C-N bond concentration (15.20 at.%) during the 5 min plasma treatment corresponded to the highest value of the surface energy (76.53 mN/m). The increases in the plasma processing time up to 30 min resulted in decreases in the atomic -C-N bond concentration and free surface energy. However, these values (12.59 at.% and 62.18 mN/m, respectively) still exceeded the initial values many times (Figure 12). 

Figure 12 shows that an increase in the total free surface energy of PLA is accompanied by an increase in the polar component (strong interactions of surface atoms with adsorbed liquid molecules and hydrogen bonds) and a simultaneous decrease in the dispersion part (van der Waals forces and other nonspecific interactions). An increase in the free surface energy of the material indicates an improvement in its adhesive properties.

Let us suppose that the -C-N bond formation in plasma treatment conditions positively influences cell viability (Figure 13). Moreover, wettability improvement, as well as increases in the specific values of the surface energy (~70–80 mN/m) [26], is also known to promote better cell survival. In our case, the PLA + N_2_ plasma sample subjected to the 5 min treatment had the optimal value of free surface energy (76.53 mN/m) and was characterized by the highest values of cell viability (Figure 9).

Changes in the microrelief (Figure 8, Table 5) because of plasma treatment affected the values of the contact angle and surface energy. It was shown that after 5 min of plasma treatment, the width of the fibers increased by approximately 5.5 times, and the roughness increased, while the highest value of surface energy was achieved. This was probably due to an increase in the “effective” surface area in contact with the liquid drop. However, a further increase in the processing time (from 10 min and more) and, therefore, an increase in the surface roughness did not promote further growth in the surface energy. Deformation of the polymer fibers occurred, the surface layer of the PLA was gradually destroyed and degraded, and its adhesive properties were reduced. The values of the surface energy somewhat decreased but remained higher than the initial value of the sample without plasma treatment. 

## 5. Conclusions

We explored the effect of nitrogen low-temperature arc discharge plasma treatment lasting for 5, 10, 20, and 30 min on the physicochemical properties and biocompatibility of PLA-based scaffolds. We found that the redistribution of the main PLA bonds, as well as -C-N bond formation, and surface morphology changes promoted the wettability improvement. The PLA surface became hydro- and oleophilic, and the values of the free surface energy reached the optimum level (~70–80 mN/m) for cell viability. The viability of macrophages exceeded the control level in the presence of the PLA scaffold treated with plasma for 5 min. Such an influence can be explained by several factors: the highest content of -C-N bonds and the optimal value of the surface energy ensured stable cell adhesion and satisfactory conditions for cell growth.

## Figures and Tables

**Figure 1 polymers-15-03381-f001:**
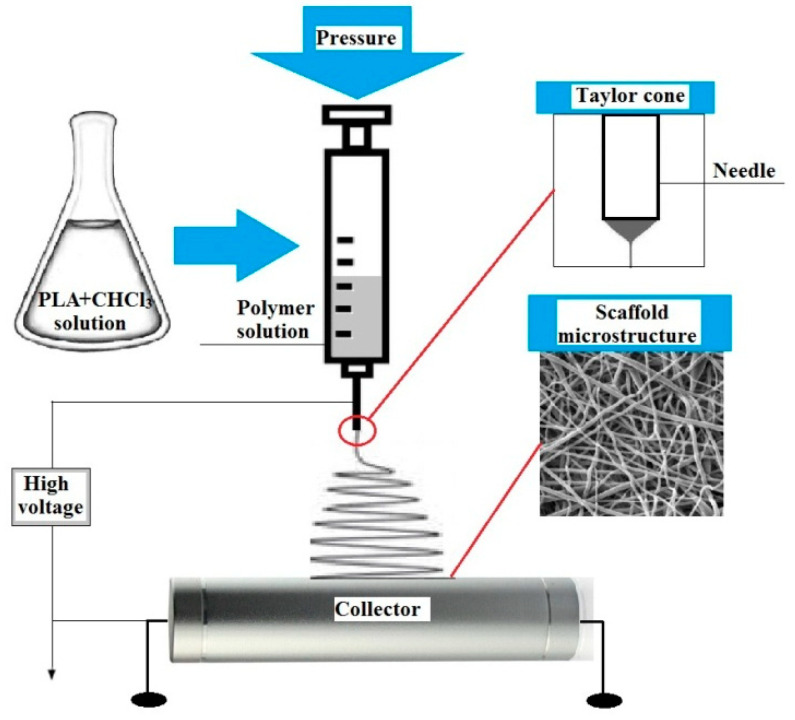
Schematic of the polymer electrospinning process.

**Figure 2 polymers-15-03381-f002:**
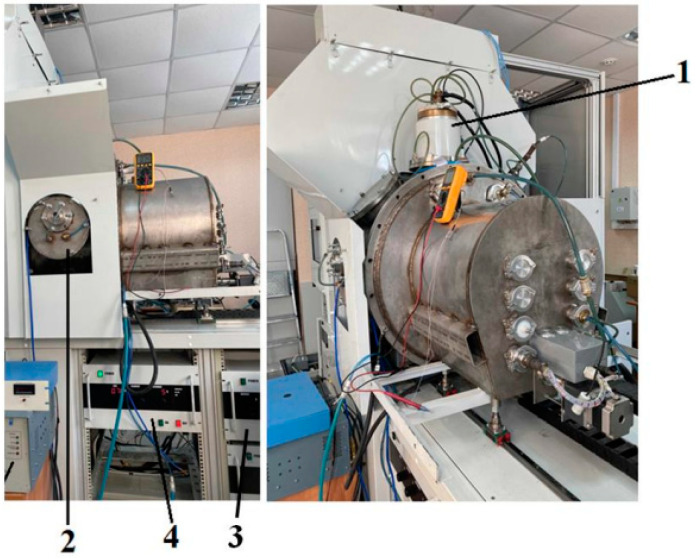
PINK plasma generator as part of the COMPLEX laboratory setup: 1—PINK plasma generator; 2—chamber for ion plasma processing; 3—electrical bias source; and 4—power supply for the PINK plasma generator.

**Figure 3 polymers-15-03381-f003:**
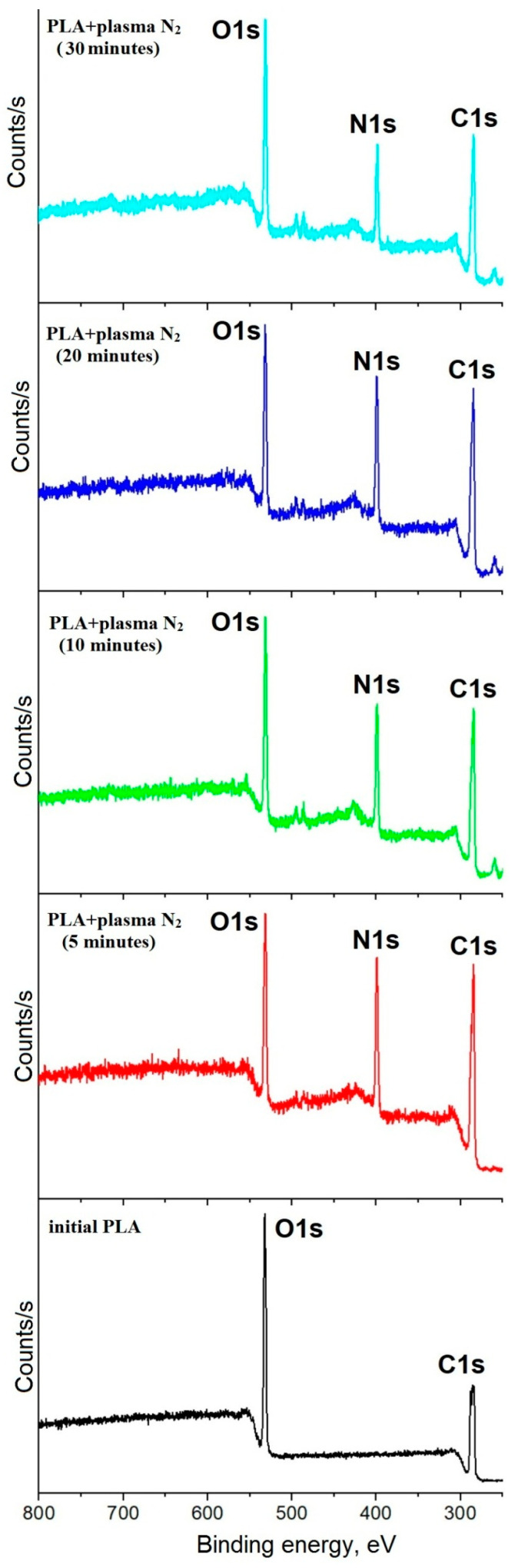
Survey XPS spectra of PLA before and after nitrogen plasma treatment.

**Figure 4 polymers-15-03381-f004:**
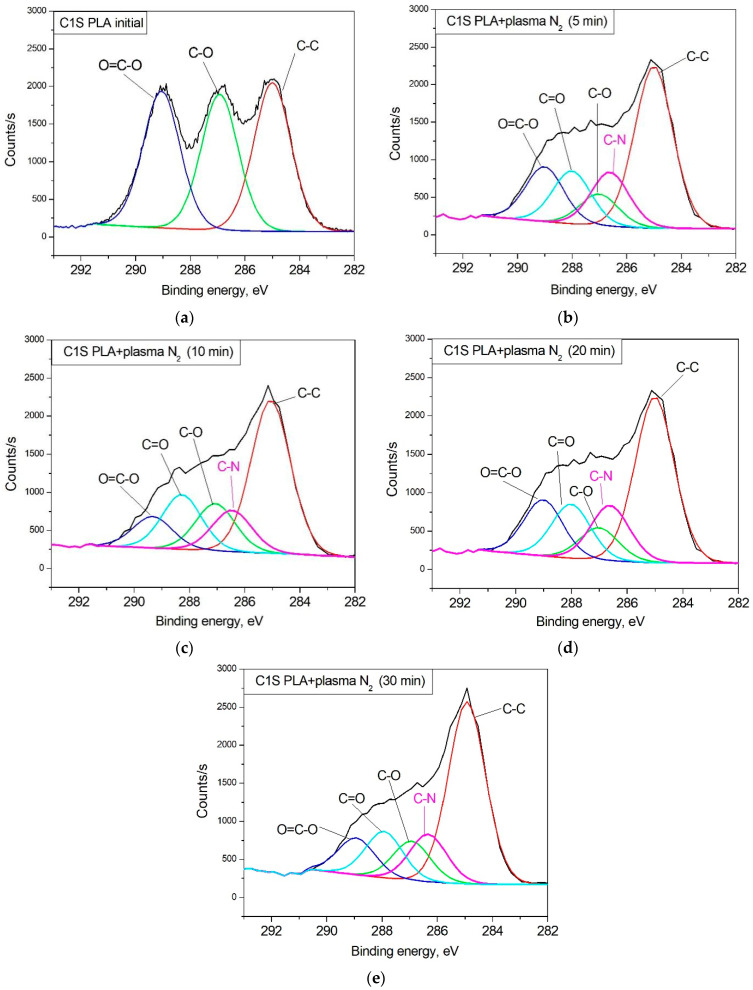
C1s XPS spectra (**a**) of the initial PLA and (**b**) PLA after nitrogen plasma treatment for 5 min; (**c**) for 10 min; (**d**) for 20 min; and (**e**) for 30 min.

**Figure 5 polymers-15-03381-f005:**
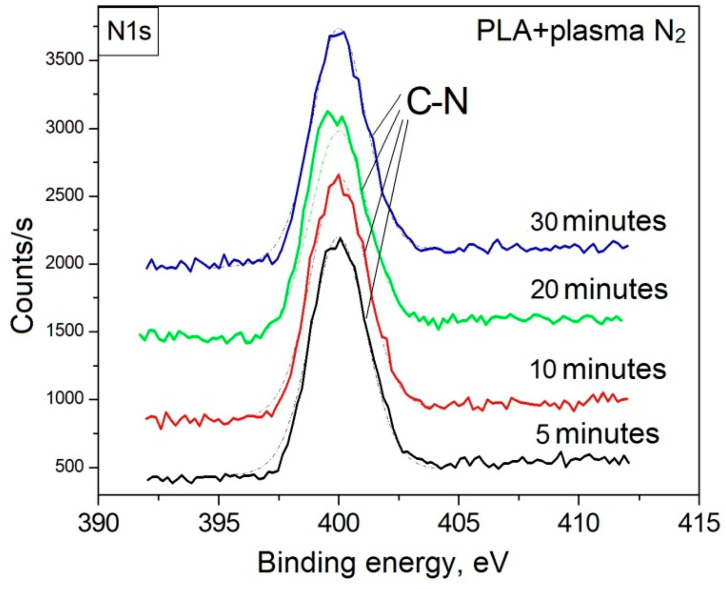
N1s XPS-spectra of the initial PLA and after nitrogen plasma treatment.

**Figure 6 polymers-15-03381-f006:**
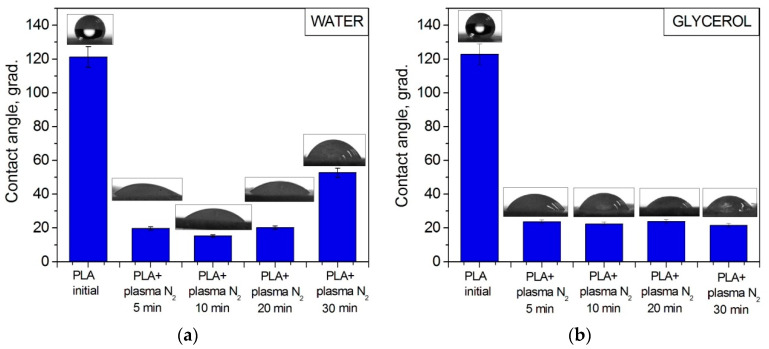
Contact angles of PLA scaffolds before and after nitrogen plasma treatment wetted with (**a**) water and (**b**) glycerol.

**Figure 7 polymers-15-03381-f007:**
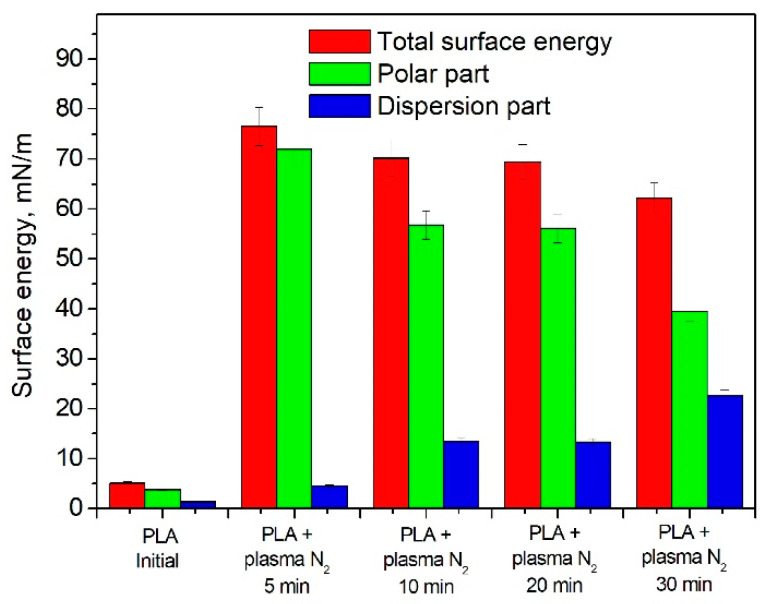
Surface energy dependence on the nitrogen plasma treatment duration.

**Figure 8 polymers-15-03381-f008:**
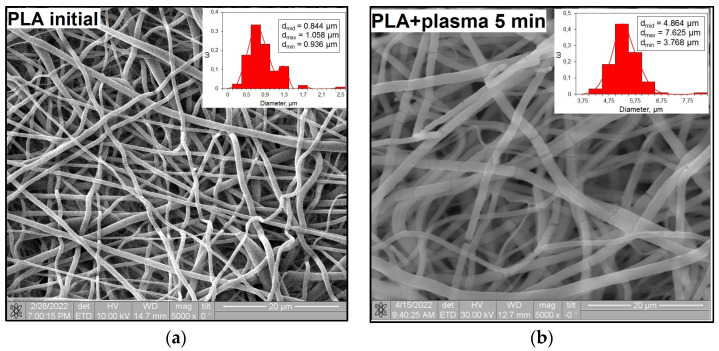
SEM images of (**a**) the initial PLA and PLA after nitrogen plasma treatment (**b**) for 5 min; (**c**) for 10 min; (**d**) for 20 min; and (**e**) for 30 min.

**Figure 9 polymers-15-03381-f009:**
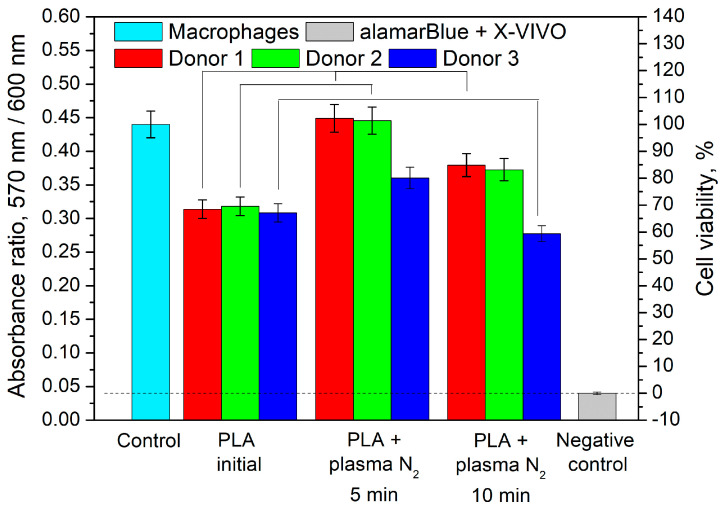
Viability of primary monocyte macrophages in the presence of the initial and plasma-modified PLA scaffolds. Differences are shown at *p* < 0.05.

**Figure 10 polymers-15-03381-f010:**
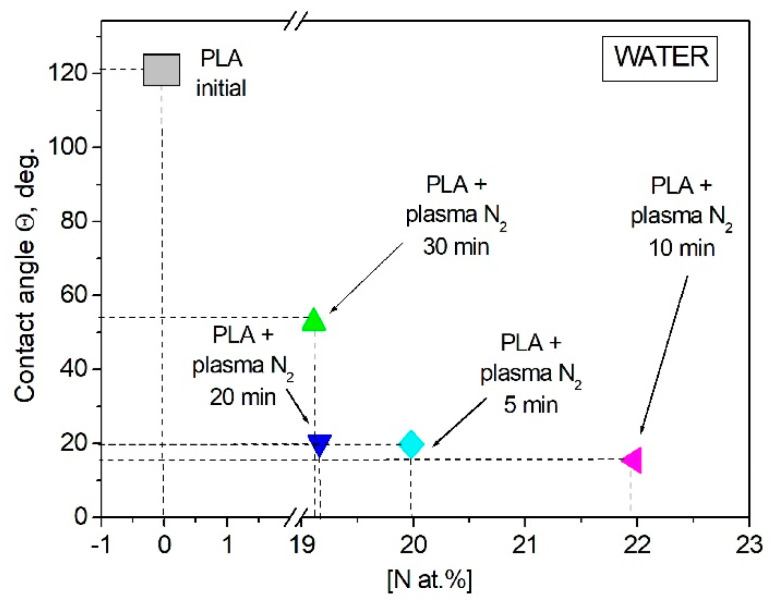
Water contact angle dependence on the nitrogen atomic concentration in the PLA surface layer.

**Figure 11 polymers-15-03381-f011:**
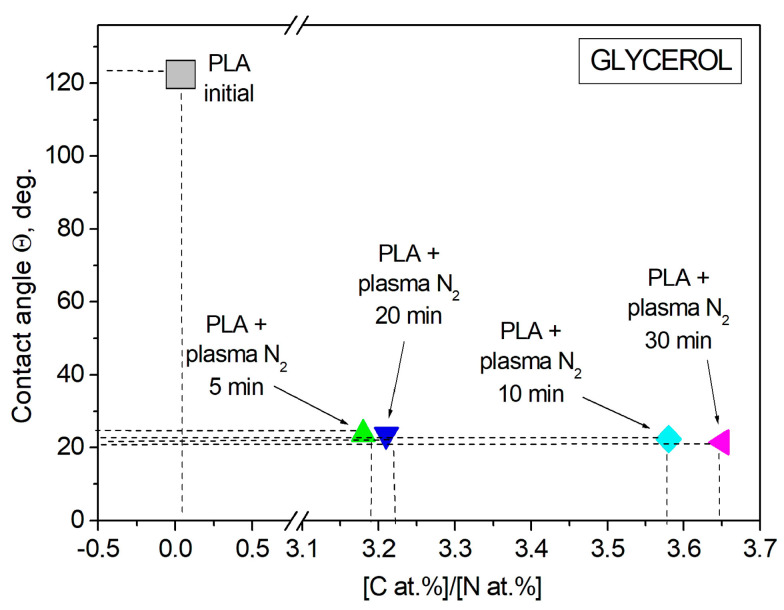
Glycerol contact angle dependence on the [C at.%]/[N at.%] ratio in the PLA surface layer.

**Figure 12 polymers-15-03381-f012:**
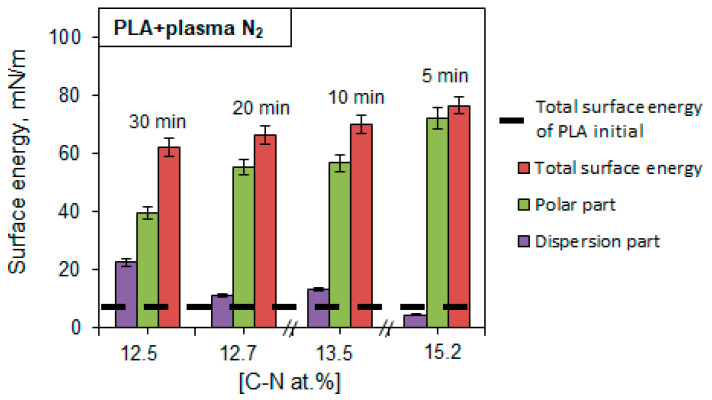
Dependence of the PLA scaffold surface energy on the -C-N bond fraction in the PLA surface layer after nitrogen plasma treatment.

**Figure 13 polymers-15-03381-f013:**
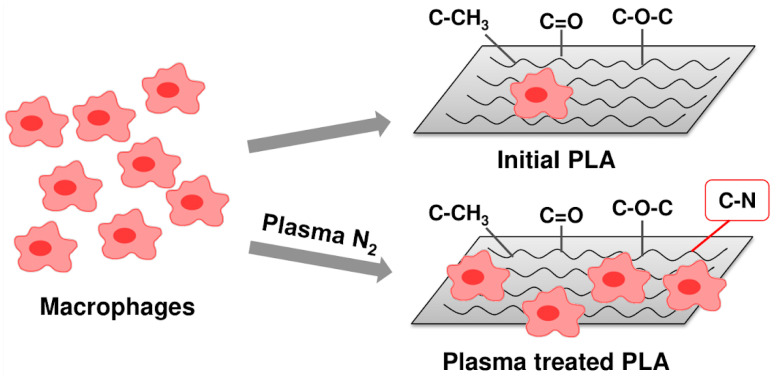
Viability of macrophages on the surface of nitrogen-plasma-modified PLA.

**Table 1 polymers-15-03381-t001:** Conditions for preparing the PLA scaffold.

Forming voltage	20 kV
Solution feed rate	3 mL/h
Solution volume	10 mL
Scaffold width	50 mm
Collector rotation	50 rpm
Distance between the needle and collector	130 mm
Needle diameter	1.3 mm (18 G)
Frequency/needle cleaning interval	10/0 min

**Table 2 polymers-15-03381-t002:** Conditions of arc discharge plasma treatment.

Plasma-forming gas			N_2_	
Pressure, Pa			0.3	
Treatment duration, min	5	10	20	30
Discharge current, A	5	5	5	5
Temperature in the chamber, °C	35	46	59	80

**Table 3 polymers-15-03381-t003:** Atomic [C, at.%]/[O, at.%] and [C, at.%]/[N, at.%] ratios of PLA before and after plasma treatment.

Sample	Ratio of Atomic Concentrations		Atomic Content of N, at.%
	[C, at.%]/[O, at.%]	[C, at.%]/[N, at.%]	
PLA initial	2.00 ± 0.2	-	-
PLA + N_2_ plasma 5 min	2.24 ± 0.2	3.21 ± 0.3	21.97 ± 0.2
PLA + N_2_ plasma 10 min	2.27 ± 0.1	3.58 ± 0.2	25.15 ± 0.3
PLA + N_2_ plasma 20 min	2.36 ± 0.5	3.18 ± 0.4	19.16 ± 0.3
PLA + N_2_ plasma 30 min	2.37 ± 0.4	3.65 ± 0.1	19.11 ± 0.2

**Table 4 polymers-15-03381-t004:** Positions of C1s lines and contents of bonds in PLA.

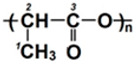		Content of Bonds in C1s Spectrum, at.%				
		CH_3_-CH- (1)	-O-CH- (2)	-O-C=O (3)	-C=O	-C-N
Sample		285.00	286.98	289.06	288.00	286.40
	Binding Energy, eV					
PLA initial		35.3 ± 0.4	32.3 ± 0.2	32.4 ± 0.4		
PLA + N_2_ plasma 5 min		45.6 ± 0.3	8.9 ± 0.2	15.5 ± 0.2	14.8 ± 0.2	15.2 ± 0.7
PLA + N_2_ plasma 10 min		51.6 ± 0.2	11.2 ± 0.2	10.5 ± 0.2	13.2 ± 0.4	13.5 ± 0.3
PLA + N_2_ plasma 20 min		47.8 ± 0.1	11.9 ± 0.2	14.1 ± 0.2	13.5 ± 0.5	12.7 ± 0.4
PLA + N_2_ plasma 30 min		46.7 ± 0.1	14.4 ± 0.4	9.6 ± 0.3	16.7 ± 0.3	12.6 ± 0.2

**Table 5 polymers-15-03381-t005:** AFM data of PLA: surface roughness and 3D images.

Sample	Surface Roughness *R_a_*, μm	3D Images of PLA Surface
PLA initial	15 ± 3	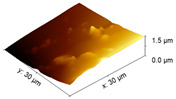
PLA + N_2_ plasma 5 min.	129 ± 23	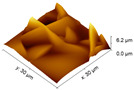
PLA + N_2_ plasma 10 min.	152 ± 27	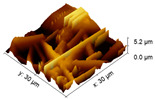
PLA + N_2_ plasma 20 min.	211 ± 37	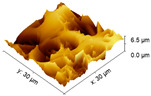
PLA + N_2_ plasma 30 min.	248 ± 41	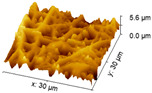

## Data Availability

The data that support the findings of this study are available. All data generated or analyzed during this study are included in this published article.

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
