# Peer review of "Effect of Nitrogen Arc Discharge Plasma Treatment on Physicochemical Properties and Biocompatibility of PLA-Based Scaffolds"

_polymers, 2023, doi:10.3390/polym15163381_

Round 1

Reviewer 1 Report

In the manuscript by Laput, the effect of nitrogen arc discharge plasma treatment on the physicochemical properties and cellular behaviors was studied. PLA scaffolds fabricated by electrospinning were exposed to low-temperature plasma treatment in a nitrogen atmosphere. The physicochemical properties of the modified surface were systematically studied, and their association with cellular behaviors was analyzed. The writing is clear, and the results are convincing. Several comments can be addressed to improve the quality.

1.      The general influences of nitrogen arc discharge plasma on the polymer can be discussed. Plasma treatment can generate heat. What is the local surface temperature during plasma treatment? The fiber can be damaged by long-time exposure. It is because of heating shock or degradation.

2.      In Figure 9, why do the negative control samples without any cells still show cell viability of 10%?

3.      The surface energy is calculated by the contact angle, which is influenced by many factors, including chemical structures and micro-topography. SEM results showed the fiber diameters and pore size change after the plasma treatment. What is the role of scaffold micro-topography change in the contact angle? The samples treated with only heating should be studied to reveal the influence of thermal-induced microstructure change.

4.      How were the cells cultured? Is it seeded on the surface of scaffolds or the well plate with the scaffold soaking in the medium? If the cell was post-seeded, typical fluorescence images for live/dead assay could be provided to show the cell adhesion.

5.      The mechanism of enhanced viability cell viability in Figure 13 is not convincing. What does the black line between the cells mean? Relevant references can be added to validate the relationship between surface energy, cell adhesion, and viability.

6.      Biocompatibility can be bio-inert or bioactive. It seems the authors refer to bioactivity based on the discussion of cell adhesion. The title can be more specific.

The writing is clear

Reviewer 2 Report

The study is interesting and relevant and warrants publication if the following issues are addressed.     

1.               The authors need to justify the uniqueness of the current study better. 

2.               More references are needed to strengthen the arguments made in the paper. The introduction needs to have more citations to the statements made.

3.       Avoiding general statements and supporting with statistics will help make the paper more relevant. The graphs like in Figure 9 have no significance analysis. It is hard to come to conclusions based on just visual graphs.

4.       The authors need to think about whether these findings were consistent in multiple trials with a bigger sample size.

5.       Further experiments to demonstrate the changes in gene or protein expression levels are needed to definitively show that the results are because of the process and not just because of primary cells adapting. The use of the Nitrogen arc has been done a while ago, and hence the authors need to probe deeper.

The authors need to use definitive language.

The punctuations and language need to be checked.

Author Response

Response to Reviewer 2 Comments

The study is interesting and relevant and warrants publication if the following issues are addressed.

  1. The authors need to justify the uniqueness of the current study better.

The literature review showed that the authors of earlier studies state the facts of changing the chemical composition, increasing the wettability of the surface, improving biocompatibility, etc.  [Ding Zh, Chen J, Gao Sh, Chang J, Zhang J, Kang ET (2004) Immobilization of chitosan onto poly-l-lactic acid film surface by plasma graft polymerization to control the morphology of fibroblast and liver cells.; Slepicka P, Trostova S, Slepickova Kasalkova N, Kolska Z, Sajdl P, Svorcık V (2012) Surface Modification of Biopolymers by Argon Plasma and Thermal Treatment.; Tverdokhlebov SI, Bolbasov EN, Shesterikov EV, Antonova LV, Golovkin AS, Matveeva VG, Petlin DG, Anissimov YG (2015) Modification of polylactic acid surface using RF plasma discharge with sputter deposition of a hydroxyapatite target for increased biocompatibility.; Techaikool P, Daranarong D, Kongsuk J et al (2017) Effects of plasma treatment on biocompatibility of poly[(L-lactide)-co-(ε-caprolactone)] and poly[(L-lactide)-co-glycolide] electrospun nanofibrous membranes.; Ayyoob M, Kim YJ (2018) Effect of chemical composition variant and oxygen plasma treatments on the wettability of PLGA thin films, synthesized by direct copolycondensation.; Yang J, Bei J, Wang Sh (2002) Improving cell affinity of poly(d, l-lactide) film modified by anhydrous ammonia plasma treatment.] and assume that these changes are interconnected with each other. The uniqueness of our study is that it shows the mechanisms of correspondence of the functional properties (wettability, biocompatibility) of surface-modified polymers to the chemical composition (atomic ratio of elements [C at.%]/[O at.%]; [C at.%]/[N at.%]). And also, there are isolated studies [Kudryavtseva V, Stankevich K, Gudima A et al (2017) Atmospheric pressure plasma assisted immobilization of hyaluronic acid on tissue engineering PLA-based scaffolds and its effect on primary human macrophages] on the effect of modified polymers on the programming of macrophages to obtain materials with predictable immunomodulatory properties.

  1. More references are needed to strengthen the arguments made in the paper. The introduction needs to have more citations to the statements made.

Thank you, it was done.

  1. Avoiding general statements and supporting with statistics will help make the paper more relevant. The graphs like in Figure 9 have no significance analysis. It is hard to come to conclusions based on just visual graphs.

Thank you for your comment.

  1. The authors need to think about whether these findings were consistent in multiple trials with a bigger sample size.

Next time we’ll undertake such a challenging study, thank you.

  1. Further experiments to demonstrate the changes in gene or protein expression levels are needed to definitively show that the results are because of the process and not just because of primary cells adapting. The use of the Nitrogen arc has been done a while ago, and hence the authors need to probe deeper.

Ok, we'll do it further. Thank you for your comment. Now we have enzyme-linked immunosorbent assay, ELISA results, but decided not to include these results in this paper.

The study of the effect of the obtained materials on the secretion of pro-inflammatory (TNFα, IL-6, IL-1β) and anti-inflammatory (IL-10) cytokines by primary human macrophages was carried out using the enzyme-linked immunosorbent assay, ELISA method and it was shown that the plasma-treated PLA scaffolds do not cause increased expression of the analyzed cytokines. Therefore, the materials do not have a pro-inflammatory, as well as an anti-inflammatory effect, and remains bioinert.

Reviewer 3 Report

The manuscript entitled “Effect of nitrogen arc discharge plasma treatment on physico-chemical properties and biocompatibility of PLA-based scaffolds” describes that low-temperature arc discharge plasma treatment in nitrogen atmosphere to influence the physiochemical property and bioactivity of PLA substrates. The method for modification is feasible for large-scale use. The manuscript was well organized and some of the problems that I have with this study are explained briefly as following:

1.     In the introduction, the authors emphasized the significance for would repair. While the scaffold for skin tissue engineering requires the suitable degradation time, bioactivity and stiffness. However, the authors did not discuss this point in whole manuscript, which might lead to the choice of materials.

2.     Although PLA possesses attractive advantages, the inflammation induced by degradation product (lactic acid) is inevitable. Please comment on this point.

3.     Normally, the measurement of surface energy is based on the contact angle of water and one/two types of organic solvents (eg. diiodomethane). The choice of glycerol is doubtful because of the high viscosity.

4.     The surface roughness of different groups was not characterized, which should be clearly discussed and is crucial for surface wettability.

5.     Macrophages were chosen for the evaluation of bioactivity. On the one hand, the relationship between macrophages and would healing was not clearly clarified. On the other hand, more tests should be performed except cell viability.

Round 2

Reviewer 2 Report

The reviewer appreciates the efforts the authors have taken to update the manuscript based on the feedback provided. However, there are a very few more changes that need to be done before the manuscript can be published.

1. The authors still have not clearly justified the rationale behind the cell culture experiments or the methodology used. The alamar blue assay uses only a 1-4hr incubation time. Why do the authors use 24 hr incubation? 

2. There are no statistical analyses done like a students t-test or ANOVA. The authors can't conclude based on averages or visually eyeing the graphs.

3.  Why were macrophages chosen? Is this to model fibrosis?

4. The novelty of this study is lacking considering similar results have been published a decade ago. The authors need to justify in detail why this is novel. Is the nitrogen arc treatment what is unique about the study?

Park, H., Lee, K.Y., Lee, S.J. et al. Plasma-treated poly(lactic-co-glycolic acid) nanofibers for tissue engineering. Macromol. Res. 15, 238–243 (2007). https://doi.org/10.1007/BF03218782

Vu, P.T., Conroy, J.P. and Yousefi, A.M., 2022. The Effect of Argon Plasma Surface Treatment on Poly (lactic-co-glycolic acid)/Collagen-Based Biomaterials for Bone Tissue Engineering. Biomimetics7(4), p.218.

Djordjevic, I., Britcher, L.G. and Kumar, S., 2008. Morphological and surface compositional changes in poly (lactide-co-glycolide) tissue engineering scaffolds upon radio frequency glow discharge plasma treatment. Applied Surface Science254(7), pp.1929-1935.

Reviewer 3 Report

The authors have mostly addressed the problems I mentioned. 

Author Response

Thank you very much!